# ESD Ideas: Photoelectrochemical carbon removal as negative emission technology

Matthias M. May[1,2] and Kira Rehfeld[3,4]

[1]Department of Chemistry, University of Cambridge, Lensfield Road, CB2 1EW, Cambridge, United Kingdom
[2]Institute for Solar Fuels, Helmholtz-Zentrum Berlin für Materialien und Energie, Hahn-Meitner Platz 1, 14109 Berlin, Germany
[3]British Antarctic Survey, High Cross, Madingley Road, CB3 0ET, Cambridge, UK
[4]Institute of Environmental Physics, Ruprecht-Karls-Universität Heidelberg, Im Neuenheimer Feld 229, 69120 Heidelberg, Germany
**Correspondence:** MM May (matthias.may@physik.hu-berlin.de)

**Abstract.** The pace of the transition to a low-carbon economy – especially in the fuels sector – is not high enough to achieve the 2°C target limit for global warming by only cutting emissions. Most political roadmaps to tackle global warming implicitly rely on the timely availability of mature negative emission technologies, which actively invest energy to remove $CO_2$ from the atmosphere and store it permanently. The models used as a basis for decarbonisation policies typically assume an imple-

mentation of such large-scale negative emission technologies starting around the year 2030, ramped up to cause net negative emissions in the second half of the century and balancing earlier $CO_2$ release. On average, a contribution of -10 Gt $CO_2$/year is expected by 2050 (Anderson and Peters, 2016). A viable approach for negative emissions should *(i)* rely on a scalable and sustainable source of energy (solar), *(ii)* result in a safely storable product, *(iii)* be highly efficient in terms of water and energy use, to reduce the required land area and competition with water and food demands of a growing world population and *(iv)*

feature large-scale feasibility and affordability.

Processes for the extraction of $CO_2$ from the atmosphere are energy-intensive. This energy has to be supplied by low- or zero-carbon sources. At present, primarily direct air capture (followed by geologic injection) and biomass production are explored and there is an active discussion on costs and scalability of the various technologies, see (Smith et al., 2016) and references cited

therein. Renewably driven direct air capture is believed to be expensive and has not yet demonstrated scalability. Therefore, the currently most feasible option appears to be the use of natural photosynthesis to generate biomass through afforestation or ocean fertilization (Smith et al., 2016). Grown plants are then permanently stored, building a new stock of fossil fuels. Alternatively, the plant material can be combusted with carbon capture and storage to act as a low-carbon fuel. However, the efficiency of natural photosynthesis drops at high light conditions and because a significant fraction of the energy is used

for the metabolism (Melis, 2012), the storage of solar energy in biomass is limited to 2-3% efficiency. Therefore large areas of agricultural land would be required for the achievement of the negative emission goals: The removal of one Gt $CO_2$/year can demand more than 1 million km$^2$, the combined area of Germany and France (Smith et al., 2016). There is an ongoing

discussion whether scaling biomass production to the required 10 Gt $CO_2$/year is at all compatible with planetary constraints (Heck et al., 2018).

We suggest to employ photoelectrochemical $CO_2$ reduction, also called *artificial photosynthesis*, to this end. As in its natural counterpart occurring in plants, photons in the artificial photosynthesis process excite charge-carriers, which then reduce (and oxidise) reactants in a liquid electrolyte to solar fuels. The photon energy is only briefly converted to electron energy, and then stored in molecular bonds. Light is absorbed in synthetic materials such as semiconductors or dyes and the chemical conversion typically takes place at (co)catalysts at the interface between electrolyte and light absorber. We primarily focus on tightly integrated photoelectrochemical systems, where the absorber is immersed into the electrolyte. While this approach imposes restrictions on the light absorber design, the tight integration also promises cost benefits (Kirner et al., 2016).

Artificial photosynthesis already delivers fivefold higher efficiencies than natural photosynthesis, as 13% for $CO_2$ reduction and 19% for solar water splitting have recently been demonstrated (Schreier et al., 2017; Cheng et al., 2018), more than half of the theoretical limits. Using solar fuels, either hydrogen or carbon-based from $CO_2$ reduction, would cut greenhouse gas emissions. However, while the combustion product of hydrogen is water, using renewably generated carbon-based fuels releases the captured greenhouse gas back into the atmosphere. Recapturing the $CO_2$ from the atmosphere would be energy-intensive and hereby lower the overall carbon reduction efficiency, which is why solar energy-driven water splitting may be the preferable fuel, eliminating carbon completely from the energy system (Parkinson, 2016). Photoelectrochemical $CO_2$ reduction could, therefore, be better placed to generate carbon-rich products that can safely and permanently store carbon extracted from the atmosphere. The electrochemical reactions have to be chosen to generate products that can be stored safely below-ground over thousands of years. Liquids or solids appear preferable, as gaseous products could leak back to the atmosphere, depending on the trapping mechanism (Damen et al., 2006). The handling of a solid product in an efficient flow-cell reactor is not practical. Although a large variety of products is in principle feasible, the production of carbon-rich liquids, such as alcohols or (fatty) acids, appears most promising. These could be stored in underground reservoirs such as depleted oil fields, be sequestered in the form of organic minerals, but also used as precursors for organic construction materials.

Any competitive artificial approach should provide a significantly higher turnover than natural photosynthesis. To assess the technologies, their efficiency for carbon removal has to be estimated and compared. The typically used solar-to-fuel efficiency (May et al., 2017) is not suitable, as it only describes the relative fraction of incident solar radiation that is converted to chemical energy. Instead, negative emission technologies based on solar energy are better assessed by the *solar-to-carbon* (STC) efficiency, which we define as the ratio of converted $CO_2$ molecules to the incoming photon flux (Appendix A).

Our calculations in the following were performed under the – highly idealised – assumption that the overpotential is dominated by the oxygen evolution reaction for a very good catalyst, which can be justified for water splitting. $CO_2$ reduction with the currently available catalysts, on the other hand, is associated with significantly higher overpotentials. The direct impact of catalysis performance on achievable efficiencies can be seen in Figure 1(a-b), where obtainable STC efficiencies and resulting module areas are plotted as a function of Tafel slope and exchange current density.

Artificial solar energy conversion does – unlike natural photosynthesis – not suffer from an efficiency decrease due to high light conditions, as beneficial effects of light concentration on the solar cell and higher temperatures on catalysis can

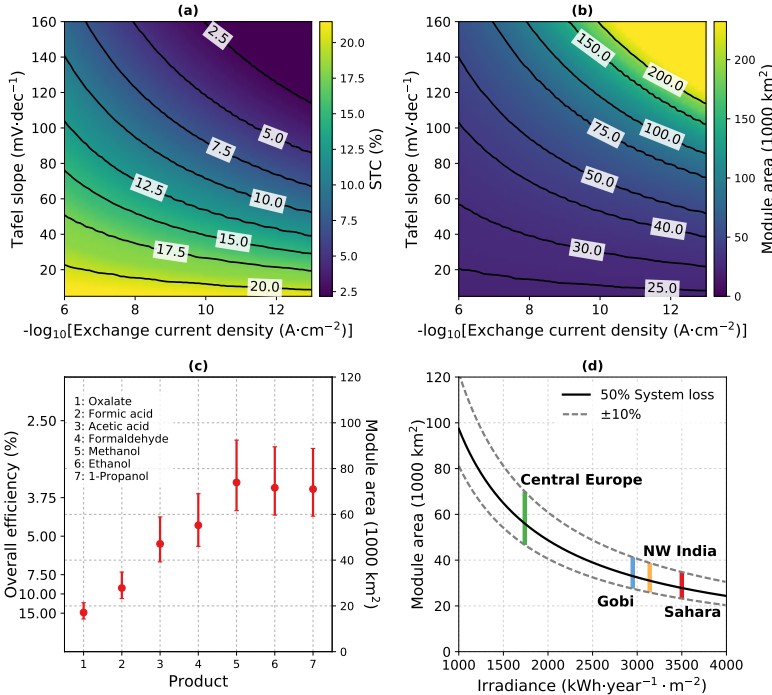

**Figure 1.** Theoretical efficiency limits and module area for the -10 Gt $CO_2$/year scenario. (a) STC efficiency limit of a dual-junction absorber for formic acid (without system loss) as a function of exchange current density and Tafel slope. (b) Resulting module area at Sahara irradiance and 50% system loss. (c) STC efficiency and module area required under Sahara irradiance for a selection of products at 50% system loss. Error bars indicate 40 and 60% loss, respectively. (d) Module area for formic acid production over the yearly irradiance at 50% (solid line), as well as 40 and 60% (dashed lines) system loss. Vertical lines mark typical irradiances accessible to a 2-axis tracker.

overcompensate the detrimental effect of temperature on the absorber. Hence, near-equatorial regions with high solar irradiation are viable target areas for its deployment. Under the assumption of 3500 kWh/m$^2$ available per year for a 2-axis tracker in the Sahara desert region (Amillo et al., 2014), we can estimate the required module area for the 2050 negative emission target of 10 Gt $CO_2$/year. At a maximum STC efficiency of ca. 19% (for formic acid, see Appendix A), this would be approximately

5    13,500 km$^2$. Under the assumption that for a mature technology the overall system efficiency is half of the theoretical efficiency, this translates to an areal requirement of about 27,000 km$^2$ (Fig. 1c). The typical space factor for tracking photovoltaics of 0.2 (Araki et al., 2016) finally leads to a land footprint of ca. 135,000 km$^2$. Other desert areas such as the Gobi desert, or the Thar desert in north-western India, would also be interesting regions. In areas such as central Europe, a lower irradiance translates to larger footprints (Fig. 1d). The scale of such an effort, if one tried to realize it in a single project, would be considerable.

10    However, it could be realized alongside with biomass approaches in other world regions, as it does not rely on agriculturally usable land. With the 2°C target, there is a truly global incentive to realise such an undertaking. Especially if spread over several projects, the economic added value would be created in the regions that suffer most under global warming.

Carbon removal by artificial photosynthesis is water-efficient, compared to its natural counterpart, as water is only used as chemical precursor and not evaporated from the closed system. Considering formic acid as product to be stored, and the target of 10 Gt $CO_2$/year to be removed, the water demand is about 4.1 Gt/year. This would be a substantial amount in dry regions. Desalination of seawater would be possible, albeit energetically inconvenient. However, the direct use of seawater was

already demonstrated for electrochemical hydrogen production (Fukuzumi et al., 2017), and might therefore also be possible for $CO_2$ reduction. Another challenge is that high-efficiency carbon sinks concentrated in large-scale facilities could, in principle, suffer from mass transport limitations of dilute $CO_2$ in the atmosphere. This could be alleviated by selecting sites with high atmospheric convection rates, by spacing facilities sufficiently widely apart, or to combine them with solar updraft towers for electricity generation.

Requirements for the safe storage will vary significantly with the choice of the sink product. Formic acid would certainly be problematic due to its corrosiveness, also in the case of spilling events. Acetic acid and the alcohols are inflammable at high concentrations and would have to be diluted with water, increasing the water and volume footprint. For oxalate, the sink product with the highest STC efficiency, mineral trapping by reaction with a suitable calcium source, such as calcium chloride, to the stable mineral whewellite could be anticipated. The 10 Gt $CO_2$ goal would result in roughly 17 Gt of the mineral. As

a solid product, this – or other organic minerals – would not require underground injection and hence at this stage appears to be the most attractive option, with similar requirements as biomass storage. However, post-processing will increase the energy footprint and hereby also the costs.

In principle, electrochemical reduction of $CO_2$ would also be possible using photovoltaics or wind power to first generate electricity, and then drive electrolysis and the chemical conversion. This introduces the intermediate step of converting solar to

electrical energy. For the scales required, it appears that the potential of solar energy will, unlike wind, not be a limiting factor (Kleidon et al., 2016). Hybrid approaches, where inorganic solar cells are combined with bacteria, are also possible (Liu et al., 2015), but efficiencies are currently low and it is unclear how the drop in production rate under high illumination conditions can be overcome.

Artificial photosynthesis in the form of $CO_2$ reduction represents consequently an interesting technological option for neg-

ative emissions due to its high efficiency. This would greatly reduce land use for the anticipated 2050 negative emission target compared to so far considered – mainly biomass-based – technologies. The installation of the required minimum module area of about 30,000 km$^2$ would, however, still be an enormous undertaking. While we estimate the costs for photoelectrochemical $CO_2$ conversion to the sink product to roughly 65 Eur per ton (see appendix B), we emphasize that the development stage of highly efficient photoelectrochemical $CO_2$ conversion does not yet allow a robust estimate of the costs, rendering this value

rather speculative. Furthermore, some of the anticipated sink products have an economic value as energy carriers and therefore require the creation of incentives to actually sequester and not combust the product. Physical feasibility and technological challenges can, however, already be anticipated. The greatest challenges to overcome towards the application are, for now, to develop and implement systems that are stable under operating conditions, as well as the derivation of earth-abundant, efficient catalysts (Schreier et al., 2017; Cheng et al., 2018).

*Code availability.* Calculations were performed with YaSoFo, the source code is available under https://bitbucket.org/YaSoFo/. The python source code to reproduce the calculations is available under DOI:10.5281/zenodo.1489158.

## Appendix A: Solar-to-carbon efficiency measure

Given a PEC device and a target sink product, we define the STC efficiency by the ratio of carbon atoms, which are chemically
fixed, over the total incoming photon flux, $j_{ph}$, given by the integrated solar spectrum. The electronic current corresponding to this total photon flux would be the photocurrent that could be extracted from an ideal absorber with an infinitesimally small bandgap, where each photon contributes to one electron in the photocurrent. The STC efficiency limit for an ideal photoelectrochemical solar cell can then be calculated as follows: The Gibbs free energy difference per electron, $\Delta G$, constitutes the electrochemical load of the cell. It limits, together with the terrestrial solar spectrum, $n(\lambda)$, the electronic current density, $j_e$.
Tandem solar cells are required for high efficiencies in photoelectrochemical energy conversion as they provide high currents and sufficient voltage to drive the reaction. The current density of an ideal tandem absorber under air mass 1.5 global illumination with very good catalysts can be calculated in the detailed-balance-scheme (May et al., 2017). Under the assumption of unity absorption above the bandgap, the top cell absorbs photons $n(\lambda)$ in the range between the far UV ($\lambda \to 0$ nm) and the wavelength corresponding to its bandgap, the bottom cell experiences the photon flux filtered by the top cell and there-
fore absorbs between the respective bandgaps of top and bottom cell. The smaller of the two values then gives the maximum photocurrent at zero load. The operational photocurrent is then obtained by intersecting the overall current-voltage curve of the solar cell with the curve of its load, given by the Gibb's Free energy of the redox couple and the catalyst characteristics described by exchange current density and Tafel slope (see Asset). The selected product then defines the electron efficiency, $\eta_e$, i.e. the inverse of how many electrons are consumed for the formation of one product molecule from $CO_2$ and water. With the
Faradaic efficiency $\eta_F$, describing the efficiency of the conversion from current to desired product, the STC can be formulated as:

$$\text{STC} = \eta_F \eta_e \min_i \left[ \frac{\int_{\lambda_i}^{\lambda_{i+1}} n(\lambda)\mathrm{d}\lambda}{\int_0^\infty n(\lambda)\mathrm{d}\lambda} \right] = \frac{\eta_F \eta_e j_e}{j_{ph}} \tag{A1}$$

For formic acid (HCOOH, $\Delta G = 1.4$ eV, $\eta_e = 0.5$), these idealised assumptions result in a maximum electronic photocurrent density of $j_e \simeq 26$ mAcm$^{-2}$. For unity Faradaic efficiency, we obtain a product current density equivalent of $\eta_e j_e =$
13 mAcm$^{-2}$. It follows that ideally ca. 19% of the incoming solar photons transform a $CO_2$ molecule to the liquid – and hence storable – product. The STC efficiency would therefore be 19%. Taking into account photoconversion, Faradaic, and system losses, values of 10% STC or more appear feasible as for the similar process of photoelectrochemical water splitting, 85% of the material-specific and ca. 2/3 of the overall theoretical efficiency limit were already demonstrated on a lab-scale (Cheng et al., 2018). This is high compared to the currently achieved energetic efficiencies for natural photosynthesis of 2-3%, which
translate to roughly 1.5-2% STC efficiency.

STC efficiencies are a function of the reaction path, similar to $CO_2$ reduction for fuel generation, where the obtainable efficiency depends on the Gibbs free energy (May et al., 2017). The distribution of energy over the chemical bonds varies for different products, yet for $CO_2$ removal, we are primarily interested in the number of converted $CO_2$ molecules. Therefore, the STC efficiencies can deviate significantly for products that have a similar energetic efficiency (Fig. 1c). Feasible products are associated with distinct storage requirements, but also different catalysts. Though the electronic photocurrent could be higher for acetic acid compared to formic acid due to a reduced electrochemical load, 4 electrons are required for the conversion of one $CO_2$ molecule, which in the end almost halves the efficiency. The theoretical efficiency limit, as shown in Figure 1c, largely varies based on the number of electrons consumed per $CO_2$ molecule, which is one for oxalate, two for formic acid, four for acetic acid and formaldehyde, and six for methanol, ethanol, and 1-propanol. Therefore, using the carbon conversion rate as the benchmark for solar-driven negative emissions will result in a different choice of product compared to solar fuels, where energetic considerations dominate.

## Appendix B: Cost estimate

To roughly estimate the costs of negative emissions by photoelectrochemical $CO_2$ reduction, we assume the module costs to be twice the module costs (Chang et al., 2018) of current crystalline silicon photovoltaics. With a depreciation period of 20 years, and running costs of 10% of the investment sum, this would translate to 55.60 Eur per ton of $CO_2$. Additional costs can arise from the diffusion limitation due to the high conversion rate, which might necessitate to technically create convection by means of mechanical fans. The energy costs of capturing atmospheric $CO_2$ are estimated to be about 30-88 kJ mol$^{-1}$ (Goeppert et al., 2012). With an average of 50 kJ mol$^{-1}$ and a current photovoltaic electricity price of 30 Eur/MWh, this adds another 9.50 Eur/ton, finally totalling 65 Eur/ton. Transport costs to the storage location will vary with the chosen product and the vicinity between production and storage facility. If we assume, as a very rough estimate, similar transport costs of formic acid as for crude oil over a distance of 2000 km (Verma et al., 2017), this would result in additional 24 Eur per ton of $CO_2$. The overall volume to be transported would be in the same order of magnitude as the present day oil production. The costs for storage will vary with the sink product, the product volume, and the required post-processing. Some of the products could, in principle, be used as 'plastic-based' construction materials, creating an economic value and hence reduce the overall costs. Considering the required scale, however, the market volume for such construction materials will probably not be significant.

*Competing interests.* There are no conflicts to declare.

*Acknowledgements.* We thank the German Research Foundation (code RE3994-1/1 and RE3994-3/1, for KR) and the fellowship programme of the German National Academy of Sciences Leopoldina, grant LPDS 2015-09 (MM) for funding. Gregor Schwerhoff, Carl Poelking, and Klaus Pfeilsticker are acknowledged for comments on the manuscript. The authors thank the referee Bruce Parkinson for the idea to use oxalate as a sink product.

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
