# Peer review of "ESD Ideas: Photoelectrochemical carbon removal as negative emission technology"

_Earth System Dynamics, 2018_

## Referee Comment (RC1) · B. Parkinson (Referee) · 31 Aug 2018

This is a totally appropriate paper for this forum. The authors take a big picture view of the need to also remove CO2 from the atmosphere as well as to stop adding it. A fact that is not being discussed as widely as it should be. The contribution is well written and well reasoned and does supply some "back of the envelope" calculations to emphasize their points albeit with admitted optimistic assumptions. I have several comments and suggestions:

1. I do find it unlikely that anyone who is investing substantial energy in a CO2 reduction product, that also has fuel value, can be persuaded to sequester the product. Even the

first reduced product, formate, that has the highest potential for efficiency is essentially a hydrogen carrier.

2. On page 2 line 19 they suggest that the reduced products could be used as precursors to "organic construction materials". Given the scale of what would be needed it would be unlikely that such a huge market for such a material would exist. And if so it would have to be something much cheaper than could be justified from the cost of the electrons needed to reduce the CO2. Large subsidies might be needed to make this possible.

3. I would like to see the authors include oxalate in their analysis since this requires only one electron per CO2 and could be stored as solid oxalic acid dihydrate by concentrating the reduced brine or as precipitated calcium oxalate if a calcium source other than CaCO3 or limestone could be used such as CaCl2 from the ocean or other natural brines.

---

## Author Comment (AC1) · 2 Sep 2018

We thank Prof. Parkinson for his valuable comments. In the following, we address them on a point-to-point basis.

**Referee:** 1. "I do find it unlikely that anyone who is investing substantial energy in a $CO_2$ reduction product, that also has fuel value, can be persuaded to sequester the product. Even the first reduced product, formate, that has the highest potential for efficiency is essentially a hydrogen carrier."

**Authors:** This concern is also valid for other negative emission approaches such as

biomass production. Therefore, the creation of incentives to actually sequester the products of negative emission technologies is certainly an issue that has to be addressed in future climate treaties. We will add a general statement to the manuscript.

**Referee:** 2. "On page 2 line 19 they suggest that the reduced products could be used as precursors to "organic construction materials". Given the scale of what would be needed it would be unlikely that such a huge market for such a material would exist. And if so it would have to be something much cheaper than could be justified from the cost of the electrons needed to reduce the $CO_2$. Large subsidies might be needed to make this possible."

**Authors:** It is true that the market for organic construction materials or other plastics will probably be smaller than the volume of products generated by negative emission technologies in the anticipated scenarios, i.e. 10 Gt $CO_2$ per year by 2050. We see this rather as an added benefit since energy and money will have to be invested anyway for the achievement of a (sub) 2°C target limit for global warming with the aid of negative emissions. To what extent the further use of the carbon-rich products (such as for organic plastics) will make sense economically, will depend on the costs of sequestration as well as the future costs for the extraction of the remaining fossil oil ressources.

**Referee:** 3. "I would like to see the authors include oxalate in their analysis since this requires only one electron per $CO_2$ and could be stored as solid oxalic acid dihydrate by concentrating the reduced brine or as precipitated calcium oxalate if a calcium source other than $CaCO_3$ or limestone could be used such as $CaCl_2$ from the ocean or other natural brines."

**Authors:** We appreciate this suggestion. Oxalate is indeed a highly interesting product which we had not included. Oxalate production from $CO_2$ comes with an electrochemical load of 1.82 eV [e.g. Qiao et al, Chem. Soc. Rev., 2014, 43, 631] and one electron per $CO_2$ molecule. The overall resulting solar-to-carbon efficiency in our model (with its idealised catalysis), would be with system loss about 15%, which is significantly higher

than formate (see Fig. 1, update to Fig. 1c,d in the manuscript). With respect to point 1, one could argue that oxalate is a lower-value molecule, which results in a higher incentive to actually sequester the product.

We will include the dicussion above and the updated figure in the revised version of our manuscript.

——————————————————————

[Figure]

**Fig. 1.** Efficiency and module area for the -10\,Gt CO2/year scenario. (a) STC efficiency and module area required under Sahara irradiance at 50% system loss. (b) Module area for formic acid production.

---

## Referee Comment (RC2) · Anonymous Referee #2 · 9 Nov 2018

I have read the paper Photoelectrochemical carbon removal as negative emission technology by May and Rehfeld. The paper presents an interesting idea that is well suited to ESD. Overall the paper seems reasonable enough, although there are a number of places were the authors should be more specific. There also needs to be additional discussion of several specific points in order to provide a fuller context. The introduction also has a few statements that are opinion and not based in scientific analysis and need to be modified since this is a scientific paper. The idea presented is interesting enough without putting up artificial strawmen to knock down! With these modifications the paper will be suitable for publication.

[Figure]

Specific Comments:

Lines 7-10

The last sentence of the abstract (and some associated text in the introduction) contains some assertions that are not supported by the literature. These detract from the paper overall. Specifically:

First "should (i) rely on an unlimited source of energy (solar),", contains a number of implicit assumptions. "unlimited" implies that negative emissions should be maintained indefinitely. If taken literally this would eventually decrease CO2 concentrations below preindustrial levels! Not a goal I suspect the authors intend nor has this been proposed. If we posit an eventual goal of a zero carbon energy system, then the role of negative emissions is a finite one whose role is to "draw down" atmospheric CO2 levels until a "safe" leve is reached (e.g. UNFCCC). Therefore, an unlimited source of energy is not required, just a source that is sufficient for the amount of negative emissions that might be needed.

While of course it is important that any such process "result in a safely storable product", there is no evidence for the second portion of this statement, that this needs to be liquid or solid, not gaseous. There is a related misstatement later about geologic CO2 storage since, in fact, at the depths where CO2 is injected for storage pressures are high enough that CO2 is liquid. A substantial body of literature has verified that CO2 injection at a number of sites around the world, the most well known of which is likely the Norwegian Schlipner natural gas field, has shown that CO2 injection results in stable, trapped CO2. Again, this does not detract from the idea proposed by the authors, but statements in a scientific paper do need to be scientifically accurate! There is also a rather long history of CO2 injection for enhanced oil recovery, which has also provided substantial information on sub-surface behavior of CO2 (albeit in a condition not designed for 100% long-term storage).

Line 15. "therefore, the currently most feasible option appears to be the use of natural

photosynthesis to generate biomass through afforestation or ocean fertilization. "

The authors do not appear to be aware of the literature here. First, it is not clear if ocean fertilization is feasible (experiments have not always provided support for this idea), nor are the side effects of ocean fertilization well enough understood.

The authors are deliberately ignoring geologic Carbon Capture and Storage (CCS), which is the leading option, along with reforestation, and related strategies, examined in scenarios to date. This is already being practiced today at a number of sites and there are no known technical barriers to large-scale application. While there may be legitimate concerns with this option, some of those concerns are also likely to apply to the option proposed by the authors. A more balanced discussion is needed (particularly since the authors later suggest geologic injection of the carbon-based liquids produced by their methods, which is inconsistent with the neglect of geologic CCS). A discussion of CCS and, presumably, contrast with the author's proposal needs to be added.

Page 1 Line 22. I am aware of course of the Heck et al. (2018) estimate, however much other literature does not come to the same conclusion (for example see the SSP scenarios, which are produced by models that integrate energy land and economics into a consistent projection framework). This not to discount the issue, but note that this is not a settled issue scientifically, and there are many nuances that prohibit such a blanket statement. For example, there is not one hard and fast estimate for the amount of negative emissions that are needed, this depends on the specific target and the rate that the world energy system is decarbonized.

However even taking the author's statement that 10 Gt/year of $CO_2$ is needed, residue biomass sources (e.g. rice straw, corn stover, etc.), which do not require any additional land, could supply this amount by mid-century as crop production expands globally to support glorwing population and affluence levels. There is abundant literature on this point.

Page 2 line 10. "These could be stored in underground reservoirs such as depleted

oil fields, but also used as precursors for organic construction materials." The problem with using these liquids as precursors for organic construction materials is that these materials will eventually oxide and return carbon to the atmosphere. It is possible that this could still result in a long-term sink, but at only a fraction of the initial flow. (There are a number of papers that examine the substitution of wood for concrete as a building material that can be used for references and guidance here and should be cited.)

Further, if the authors are proposing use as construction materials, than the energetic cost of converting liquids to solids should be discussed.

Page 2 Line 17. As noted above this statement is not correct. First CO2 is not a gas at the relevant pressures, and there is abundant literature and data that indicate that CO2 can be safely injected into formations where it appears to be trapped on a long-term basis. (Some of these statements may indicate a misunderstanding of how deep CO2 injection works. Injected CO2 does not form some sort of large "bubble", but instead moves into rock pore spaces where chemical and physical reactions then occur.) .

Page 3 Line 6. Reference needed for what "previously largest project for solar electricity production in the Sahara desert" since this is a moving target!

Page 4, line 10. This "Unlike solar energy, however, large-scale mechanical wind extraction from the atmosphere might be limited. (Miller et al., 2011)" is another strawman that is irrelevant. We do not need unlimited energy for this process, just sufficient, affordable energy. In some locations that might best be supplied by wind, some solar, and in others biomass.

Page 4 line 16 - "While we estimate the costs for photoelectrochemical CO2 removal to roughly 65 Eur per ton" (add reference to the appendix where the details are given)

page 4. What is missing here are a discussion of transportation and storage costs. For any system that is sufficiently large scale, pipelines would likely be the most efficient mechanism for transport. Costs of pipeline transport of CO2 are well known

(this is already a common practice) as also, of course, petroleum products, so a direct comparison of costs per km per tonne of C should be feasible and would be very useful. There are unlikely to be that many suitable storage locations in the places with the highest solar irradiance. So long-distance transport would likely be necessary. This is not necessarily a huge barrier (current scenarios with CO2 with CCS envision long-distance transport of CO2, and the world currently transports large quantities of petroleum across the globe - this would actually be a useful comparison. Are we talking about a volumetric transport rate per year that is on the order, much less than, or greater than current international shipments of oil by tanker? See the data in the "Review of Maritime Transport" series by the UNCTAD).

The volume needed for storage is a significant potential issue for this proposed negative emissions mechanism and needs to have more discussion and some order of magnitude estimates.The total storage volume over, say a 50 to 100 year period should be estimated by the authors and compared with potentially available storage locations. The authors mentioned depleted oil fields - the problem here is the same one mentioned by the authors as an issue for CCS with CO2, liquids injected into the oil fields may come to the surface (given that oil already comes to the surface in production wells).

There will also be some cost for storage. Injection into geologic formations (such as old oil wells) requires compression and pumping. The cost of this is well known and should be added to the cost estimates in Appendix B.

One advantage of CO2 sequestration is that CO2 is harmless, although it does pose a danger at concentrations large enough to exclude oxygen (e.g. this is the sort of low probability high consequence event considered by regulators in developing CO2 pipeline, and ultimately CCS, regulations). The potential health and environmental issues associated with the proposal should also be mentioned. One would presume that there could be environmental damage due to spills of these hydrocarbons from pipelines and/or leakage into groundwater or to the surface from storage sites. Are
there likely to be tradeoffs between carbon density, transportation issues (e.g. low enough viscosity to be transported by pipeline), and low toxicity or other environmental impacts?

Page 5 the statement "these idealised assumptions result in a maximum electronic photocurrent density of je âĽČ26mAcm−2" needs support. How was this value obtained. What assumptions were used to obtain this specific value.

Similarly here: "It follows that ideally ca. 19% of the incoming solar photons transform a CO2 molecule to the liquid," as the percent value does not seem obvious or related to the above values.

The values seem reasonable, but it is important to state one's assumptions so that readers know exactly what is being referred to here.

It is also important to note that these values are ideal cases. This needs to be emphasized in the main paper. This is fine, and is a reasonable starting point for bounding the possibilities, but it needs to be made clear that this is just the theoretical limit, which is unlikely to be approached commercially. (If we could approach the efficiency of nature, it would be a great starting point.)

This "$\eta e = 0.5$" seem backwards, as the variable is described as "how many electrons are consumed for the formation of one product molecule from CO2 and water", which would suggest the value should be 2, not 0.5. But perhaps the definition was not written correctly?

---

## Author Comment (AC2) · 17 Nov 2018

We thank the referee for the detailed and constructive criticism. In the following, we address the comments on a point-to-point basis. We also indicate the changes to be implemented in our manuscript.

**Referee:** 1. "The last sentence of the abstract (and some associated text in the introduction) contains some assertions that are not supported by the literature. These detract from the paper overall. Specifically:

First "should (i) rely on an unlimited source of energy (solar),", contains a number of implicit assumptions. "unlimited" implies that negative emissions should be maintained indefinitely. If taken literally this would eventually decrease $CO_2$ concentrations below preindustrial levels! Not a goal I suspect the authors intend nor has this been proposed. If we posit an eventual goal of a zero carbon energy system, then the role of negative emissions is a finite one whose role is to "draw down" atmospheric $CO_2$ levels until a "safe" leve is reached (e.g. UNFCCC). Therefore, an unlimited source of energy is not required, just a source that is sufficient for the amount of negative emissions that might be needed."

**Authors:** *We agree that a reduction of $CO_2$ concentrations below preindustrial levels is most probably not desirable. The length of time, for which negative emissions will be required, depends on the pace and the completeness of the transition towards a low- or zero-carbon economy and is hitherto unclear. By "unlimited" we did, however, not want to express "infinite", but rather "non-limiting" in a sense that the energy source should scale up to the required level and not deplete over the anticipated timeframe in the order of at least several decades or a century (as also mentioned by the Referee in point 12). The decarbonisation of the electricity sector, the fuels sector, a growing world population, increased standards of living, and finally negative emission technologies will all add to the demand for sustainable energy sources. Consequently, it is important that there are at least no physical limits that might arise from the sheer scale of the undertaking. This does, however, not exclude economic barriers.*

**Changes to the manuscript:** To avoid this misunderstanding, we will change the wording to "[...](i) rely on a scalable and sustainable source [...] ". We will furthermore correct "(iv) be [...]" to "(iv) feature [...]".

**Referee:** 2. "While of course it is important that any such process "result in a safely storable product", there is no evidence for the second portion of this statement, that this needs to be liquid or solid, not gaseous. There is a related misstatement later

about geologic $CO_2$ storage since, in fact, at the depths where $CO_2$ is injected for storage pressures are high enough that $CO_2$ is liquid. A substantial body of literature has verified that $CO_2$ injection at a number of sites around the world, the most well known of which is likely the Norwegian Schlipner natural gas field, has shown that $CO_2$ injection results in stable, trapped $CO_2$. Again, this does not detract from the idea proposed by the authors, but statements in a scientific paper do need to be scientifically accurate! There is also a rather long history of $CO_2$ injection for enhanced oil recovery, which has also provided substantial information on sub-surface behavior of $CO_2$ (albeit in a condition not designed for 100% long-term storage)."

**Authors:** *Gaseous products are, by their intrinsically more volatile nature, more challenging to contain over long periods of time than solids or liquids. The trapped $CO_2$ mentioned by the referee is in this "product" state also not gaseous, but might, upon change of the conditions (e.g. a variation in pressure) return to the gaseous state. We do, however, agree that without further elaboration, this short statement does not improve clarity.*

**Changes to the manuscript:** We will remove "(e.g. liquid or solid, not gaseous)".

**Referee:** 3. "Line 15. 'therefore, the currently most feasible option appears to be the use of natural photosynthesis to generate biomass through afforestation or ocean fertilization.'

The authors do not appear to be aware of the literature here. First, it is not clear if ocean fertilization is feasible (experiments have not always provided support for this idea), nor are the side effects of ocean fertilization well enough understood."

**Authors:** *We are aware of the literature. The compactness of the "ESD Idea" does, however, only allow us to refer the reader to select review articles, where the different technologies are discussed in more detail, e.g. the reference [Smith et al., Nature Climate Change 6, 42 (2016).] We agree that each technology has its specific advantages*

*and risks and/or limits. Consequently, at the current level of negative technologies, the judgement, which technology is most promising, depends to a large degree on the relative weight of the parameters. We rephrase some of our statements to clearly indicate that this is a lively and ongoing discussion in the community and emphasize that there is indeed abundant literature.*

**Changes to the manuscript:** We will add "[... are explored] and there is an active discussion on costs and scalability of the various technologies, see (Smith et al., 2016) and references cited therein.". Furthermore, we will repeat the Smith et al. reference at the end of our statement "[...] ocean fertilization.".

**Referee:** 4. "The authors are deliberately ignoring geologic Carbon Capture and Storage (CCS), which is the leading option, along with reforestation, and related strategies, examined in scenarios to date. This is already being practiced today at a number of sites and there are no known technical barriers to large-scale application. While there may be legitimate concerns with this option, some of those concerns are also likely to apply to the option proposed by the authors. A more balanced discussion is needed (particularly since the authors later suggest geologic injection of the carbon-based liquids produced by their methods, which is inconsistent with the neglect of geologic CCS). A discussion of CCS and, presumably, contrast with the author's proposal needs to be added."

**Authors:** *With "primarily direct air capture" in "At present, primarily direct air capture and biomass production are explored" on page one, we had geological storage of the captured $CO_2$ in mind, referring – for the sake of brevity – to the comparative review paper of Smith et al. (2016). In doing so, we followed the terminology of the aforementioned paper, where 'CCS' refers to the capture of emissions from fossil fuels, which does not lead to net negative emissions.*

**Changes to the manuscript:** We will add "followed by geologic injection" to "[...] direct

air capture (followed by geologic injection) [...]"  (For further amendments, see our comments to related point 7.)

**Referee:** 5. "Page 1 Line 22. I am aware of course of the Heck et al. (2018) estimate, however much other literature does not come to the same conclusion (for example see the SSP scenarios, which are produced by models that integrate energy land and economics into a consistent projection framework).  This not to discount the issue, but note that this is not a settled issue scientifically, and there are many nuances that prohibit such a blanket statement. For example, there is not one hard and fast estimate for the amount of negative emissions that are needed, this depends on the specific target and the rate that the world energy system is decarbonized. However even taking the author's statement that 10 Gt/year of $CO_2$ is needed, residue biomass sources (e.g. rice straw, corn stover, etc.), which do not require any additional land, could supply this amount by mid-century as crop production expands globally to support glorwing population and affluence levels. There is abundant literature on this point."

**Authors:** *This discussion is reflected in the review papers, we refer to (e.g. Smith et al., 2016; Anderson and Peters, 2016), and for the sake of brevity, we cannot dive more deeply into it. We reformulate our statement to put more emphasis on the discussion status.*

**Changes to the manuscript:** We will change this to "There is an ongoing discussion whether scaling biomass production to the required 10 Gt/year is at all compatible with planetary constraints.(Heck et al., 2018)".

**Referee:** 6. "Page 2 line 10. "These could be stored in underground reservoirs such as depleted oil fields, but also used as precursors for organic construction materials." The problem with using these liquids as precursors for organic construction materials is that these materials will eventually oxide and return carbon to the atmosphere. It is

possible that this could still result in a long-term sink, but at only a fraction of the initial flow. (There are a number of papers that examine the substitution of wood for concrete as a building material that can be used for references and guidance here and should be cited.)

Further, if the authors are proposing use as construction materials, than the energetic cost of converting liquids to solids should be discussed."

**Authors:** *We would expect the lifetime of the organic construction material to be similar to conventional plastic, i.e. decades to centuries. Due to the compactness of the 'ESD Ideas' format, we can, unfortunately, not go into a deeper discussion at this point. Also, as Referee B. Parkinson pointed out, this pathway will probably not be very relevant, as the market will be too small for the required scale.*

**Referee:** 7. "Page 2 Line 17. As noted above this statement is not correct. First $CO_2$ is not a gas at the relevant pressures, and there is abundant literature and data that indicate that $CO_2$ can be safely injected into formations where it appears to be trapped on a long-term basis. (Some of these statements may indicate a misunderstanding of how deep $CO_2$ injection works. Injected $CO_2$ does not form some sort of large "bubble", but instead moves into rock pore spaces where chemical and physical reactions then occur.) ."

**Authors:** *Again, we would like to refer to the review paper, which we are citing. The trapping mechanisms, as well as the probability of their failure, are discussed in the work of Damen et al. Interestingly, they consider the mechanism of mineral trapping as most secure, an approach that would also be feasible with the sink product oxalate (see below).*

**Changes to the manuscript:** We will rephrase "this leaves us with liquids or solids, since gases, such as $CO_2$ itself, can leak back to the atmosphere" to "Therefore, liquids or solids appear preferable, as gaseous products could leak back to the atmosphere,

depending on the trapping mechanism.(Damen et al., 2006)"

**Referee:** 8. "Page 3 Line 6. Reference needed for what "previously largest project for solar electricity production in the Sahara desert" since this is a moving target!"

**Authors:** *We had the – currently stalled – DESERTEC project in mind, but agree with the referee that the scale is a moving target and therefore omit this argument in the revised manuscript for the sake of brevity.*

**Changes to the manuscript:** We will delete "[...] and about one order of magnitude larger than the previously largest project for solar electricity production in the Sahara desert."

**Referee:** 9. "Page 4, line 10. This "Unlike solar energy, however, large-scale mechanical wind extraction from the atmosphere might be limited. (Miller et al., 2011)" is another strawman that is irrelevant. We do not need unlimited energy for this process, just sufficient, affordable energy. In some locations that might best be supplied by wind, some solar, and in others biomass."

**Authors:** *We do not believe that this is a "strawman" because it is important to consider the whole transformation process towards a carbon-free economy. That includes renewable electricity and fuels production. At the same time, agricultural production is to be maintained for a growing (both in numbers and affluence, as pointed out by the referee above) world population, and, finally, negative emissions are required to stabilize climate. All these transformations depend on energy and, ultimately, land. Consequently, efficiency and scalability are crucial for the success of the process and it is therefore important to acknowledge or discuss potential limitations of the two main "renewable energy" technologies with the currently highest growth rates, i.e. direct solar and wind power.*

**Changes to the manuscript:** We will replace the sentence with "For the scales required, it appears that the potential of solar energy will, unlike wind, not be a limiting factor.[Kleidon et al. in *Solar Energy for Fuels* (2016)]", changing the reference to a later publication of the authors, where they directly compare wind and solar energy.

**Referee:** 10. "Page 4 line 16 - "While we estimate the costs for photoelectrochemical $CO_2$ removal to roughly 65 Eur per ton" (add reference to the appendix where the details are given)."

**Authors:** *We agree.*

**Changes to the manuscript:** We will add a reference to the appendix.

**Referee:** 11. "page 4. What is missing here are a discussion of transportation and storage costs. For any system that is sufficiently large scale, pipelines would likely be the most efficient mechanism for transport. Costs of pipeline transport of $CO_2$ are well known (this is already a common practice) as also, of course, petroleum products, so a direct comparison of costs per km per tonne of C should be feasible and would be very useful. There are unlikely to be that many suitable storage locations in the places with the highest solar irradiance. So long-distance transport would likely be necessary. This is not necessarily a huge barrier (current scenarios with $CO_2$ with CCS envision long-distance transport of $CO_2$, and the world currently transports large quantities of petroleum across the globe - this would actually be a useful comparison. Are we talking about a volumetric transport rate per year that is on the order, much less than, or greater than current international shipments of oil by tanker? See the data in the "Review of Maritime Transport" series by the UNCTAD)."

**Authors:** *We appreciate this suggestion. This is a good point. We added this consideration in Appendix B.*

**Changes to the manuscript:** We will add "Transport costs to the storage location will vary with the chosen product and the vicinity between production and storage facility. If we assume, as a very rough estimate, similar transport costs of formic acid as for crude oil over a distance of 2000 km,[Verma et al., Energy 124 (2017).] this would result in additional 24 Eur per ton $CO_2$. The overall volume to be transported would be in the same order of magnitude as the present day oil production." to the end of Appendix B.

**Referee:** 12. "The volume needed for storage is a significant potential issue for this proposed negative emissions mechanism and needs to have more discussion and some order of magnitude estimates. The total storage volume over, say a 50 to 100 year period should be estimated by the authors and compared with potentially available storage locations. The authors mentioned depleted oil fields - the problem here is the same one mentioned by the authors as an issue for CCS with $CO_2$, liquids injected into the oil fields may come to the surface (given that oil already comes to the surface in production wells)."

**Authors:** *We agree that both a substantial storage volume will be required, and that this volume has to be sealed off from the surface. However, the volume depends to a large degree on the product and its further processing. A detailed estimate of this volume is beyond the scope of this manuscript. We can, however, expect that the volume is not significantly larger than that required for the application of direct air capture followed by geological injection. Some products (such as oxalate-derived minerals) could also be stable enough to be stored above-ground.*

**Changes to the manuscript:** See reply to point 13 directly below.

**Referee:** 13. "There will also be some cost for storage. Injection into geologic formations (such as old oil wells) requires compression and pumping. The cost of this is well known and should be added to the cost estimates in Appendix B."

**Authors:** *The costs for storage will vary significantly with the chosen sink product, especially if it is further processed. Cost estimates for all the different options is beyond the scope of this manuscript.*

**Changes to the manuscript:** We will add "The costs for storage will vary with the sink product, the product volume, and the required post-processing."

**Referee:** 14. "One advantage of $CO_2$ sequestration is that $CO_2$ is harmless, although it does pose a danger at concentrations large enough to exclude oxygen (e.g. this is the sort of low probability high consequence event considered by regulators in developing $CO_2$ pipeline, and ultimately CCS, regulations). The potential health and environmental issues associated with the proposal should also be mentioned. One would presume that there could be environmental damage due to spills of these hydrocarbons from pipelines and/or leakage into groundwater or to the surface from storage sites. Are there likely to be tradeoffs between carbon density, transportation issues (e.g. low enough viscosity to be transported by pipeline), and low toxicity or other environmental impacts?"

**Authors:** *This is indeed an interesting point that we can only briefly cover here as again, it varies significantly with the chosen product and its post-processing. We will add a short discussion on this.*

**Changes to the manuscript:** We will add "Related to storage are also environmental issues from potential spilling events. Here, formic acid could be problematic due to its corrosiveness. Oxalate, on the other hand, could be processed further with calcite to the stable mineral weddellite." to Appendix A after "[...] because of its high melting point.".

**Referee:** 15. "Page 5 the statement "these idealised assumptions result in a maximum electronic photocurrent density of je mAcm$^{-2}$" needs support. How was this value

obtained. What assumptions were used to obtain this specific value.

Similarly here: "It follows that ideally ca. 19% of the incoming solar photons transform a $CO_2$ molecule to the liquid," as the percent value does not seem obvious or related to the above values.

The values seem reasonable, but it is important to state one's assumptions so that readers know exactly what is being referred to here."

**Authors:** *We expand our description of the calculation. The full calculations are accessible by inspecting the publicly available source code of the solar fuel calculator YaSoFo and the Python notebook, we will make both accessible as asset with our manuscript.*

**Changes to the manuscript:** In the appendix A, we will insert "total" to "[...] over the total incoming photon flux, $j_{ph}$, given by the integrated solar spectrum." and will add thereafter the sentence "The electronic current corresponding to this total photon flux would be the photocurrent that could be extracted from an ideal absorber with an infinitesimally small bandgap, where each photon contributes to one electron in the photocurrent." Furthermore, we added "under air mass 1.5 global illumination" to "The current density of an ideal tandem absorber under air mass 1.5 global illumination with [...]" and thereafter "Under the assumption of unity absorption above the bandgap, the top cell absorbs photons $n(\lambda)$ between $\lambda \to 0$ nm and the wavelength corresponding to its bandgap, the bottom cell experiences the photon flux filtered by the top cell and therefore absorbs between the respective bandgaps of top and bottom cell. The smaller of the two values then gives the maximum photocurrent at zero load. The operational photocurrent is then obtained by intersecting the overall current-voltage curve of the solar cell with the curve of its load, given by the Gibb's Free energy of the redox couple and the catalyst characteristics described by exchange current density and Tafel slope. [See asset].".

**Referee:** 16. "It is also important to note that these values are ideal cases. This

needs to be emphasized in the main paper. This is fine, and is a reasonable starting point for bounding the possibilities, but it needs to be made clear that this is just the theoretical limit, which is unlikely to be approached commercially. (If we could approach the efficiency of nature, it would be a great starting point.)"

**Authors:** *We already stated in the introduction that 'more than half of the theoretical limits' have been achieved for the similar processes water splitting and $CO_2$ reduction for solar fuels and "highly idealised" on page 2. We acknowledge that we should make this clearer in the discussion.*

**Changes to the manuscript:** We will change "Taking into account photoconversion, Faradaic, and system losses, values of 10% STC or more appear feasible." (Appendix A) to "[... or more appear feasible] as for the similar process of photoelectrochemical water splitting, 85% of the material-specific and ca. 2/3 of the overall theoretical efficiency limit were already demonstrated on a lab-scale.[Cheng et al.]" and removed "conservative [assumption]" on page three.

**Referee:** 17. "This "$\eta_e$ = 0.5" seem backwards, as the variable is described as "how many electrons are consumed for the formation of one product molecule from $CO_2$ and water", which would suggest the value should be 2, not 0.5. But perhaps the definition was not written correctly?"

**Authors:** *This was indeed stated ambiguously. For two electrons, the electron efficiency should be 0.5, for four 0.25.*

**Changes to the manuscript:** We will change "[...]i.e. how many electrons [...]" to "[...]i.e. the inverse of how many electrons".